# Lecturer Readiness for Online Classes during the Pandemic: A Survey Research

**Kasiyah Junus** [1,*], **Harry Budi Santoso** [1], **Panca Oktavia Hadi Putra** [1], **Arfive Gandhi** [1] and **Titin Siswantining** [2]

1   Faculty of Computer Science, Universitas Indonesia, Depok 16424, Indonesia; harrybs@cs.ui.ac.id (H.B.S.); hadiputra@cs.ui.ac.id (P.O.H.P.); arfive.gandhi@ui.ac.id (A.G.)
2   Faculty of Mathematics and Natural Sciences, Universitas Indonesia, Depok 16424, Indonesia; titin@sci.ui.ac.id
*   Correspondence: kasiyah@cs.ui.ac.id

**Abstract:** Due to the COVID-19 pandemic, most educational institutions across the world have shifted their teaching and learning processes and put efforts into preparing online distance education to ensure education continues uninterrupted. Some did not face difficult tasks or challenges during this process because they were already implementing online or blended learning before the pandemic. However, some institutions, lecturers and students were not ready to adapt to the conditions, and it is therefore important to examine to what extent lecturers are ready to teach online. This research aims to evaluate the readiness of lecturers during a pandemic that arises unexpectedly. It also aims to investigate the weaknesses and obstacles that lecturers must overcome in order to teach an online class. This research applies a mixed-method approach. Lecturers were surveyed through online preparedness questionnaires, and several themes were constructed from the gathered qualitative data. The results show that lecturers have strong baseline technical skills to use e-learning platforms for online courses; they have quickly adapted to using a Learning Management System (LMS), and most have a tactical solution for most online classes with insufficient feasibility, but they do not have a strategic solution. Their sufficiency for teaching online courses was not optimised since they did not fully believe the learning goals could be achieved. This paper elaborates on the theoretical and practical implications.

**Keywords:** instructor readiness; e-learning readiness; online teaching; pandemic

## 1. Introduction

The digital era encourages the use of Information Technology (IT) in the education sector. It facilitates online classes as a manifestation of the e-learning concept and allows lecturers and students to engage in a virtual environment, although physically separated. A concrete example of IT utilisation is a Learning Management System (LMS) platform that mediates learning processes by enabling course material repositories, student activity trackers, assignment submission and review and discussion amongst participants. Online classes are a growing trend in digital transformation and are offered by many universities, but online classes are partially combined with physical classes. Moreover, many universities use an LMS only as a course-material repository and course reporting system because there is less communication and interaction amongst participants than in a physical class.

Unexpectedly, the COVID-19 virus spread, and the World Health Organization declared COVID-19 a global pandemic, which spread to more than 150 countries. It led to the closure of offices, markets, schools and all public areas [1] in an effort to minimise the spread of the virus. Universities in Indonesia ceased all physical activities, and people rushed to carry out their activities online [2]. To continue the learning process, the Republic of Indonesia's Ministry of Education instructed all institutions to switch to fully online classes as an alternative. Unfortunately, this policy was implemented without assessing

lecturers' readiness, and the government should have been more agile. There was a similar situation in other countries [3–5].

Online class implementation required a radical change by lecturers and students regarding communication style, summative assessments and content delivery. As a fundamental problem, only a few lecturers had enough experience to conduct online courses [6], especially fully online courses. However, e-learning was expected to positively impact motivation, autonomy and student participation [7]. Therefore, readiness should be considered a critical factor when determining the success of implementation [3]. In addition, the understanding of lecturers' readiness needs to be ensured [5,8] in order to guarantee that online learning is implemented successfully. Previously, Scherer et al. [9] reported the quantitative measurement of online teaching in higher education since the occurrence of the COVID-19 pandemic. Unfortunately, it did not involve Indonesia, which has various population characteristics and unequal digital literacy. In another case, Pokrovskaia et al. also identified the e-learning implementation in Russia during the COVID-19 pandemic using three hypotheses [10] which its results focused on hypotheses testing only. Those studies utilized a quantitative approach only so the respondents were limited in expressing their perception when evaluating the online classes.

This research aims to evaluate the readiness of lecturers during a pandemic that has arisen unexpectedly. This research also has the purpose to investigate the weaknesses and obstacles that lecturers must overcome in order to teach an online class when required to do so. This study is guided by the following research questions:

1. What are lecturers' readiness levels to conduct online classes during a pandemic?
2. What are lecturers' expectations when participating in online distance education?

This research considers whether all lecturers have appropriate readiness to conduct online classes. As a practical benefit, this research contributes as stakeholders' information as one of the bases to implement tactical policies that improve teaching staff readiness for online distance teaching.

## 2. Literature Review

Before discussing this study, this section briefly defines the terms readiness, preparedness and capabilities. According to the Cambridge Advanced Learner's Dictionary, readiness is the willingness to prepare, or the state of being prepared for something [11]. A term that is similar to readiness is preparedness, but experts use these terms interchangeably. The same dictionary defines preparedness as the formal state of being prepared for a situation. By this definition, preparedness includes readiness and willingness. This study assumes that preparedness and readiness are synonymous.

However, experts differentiate between preparedness and competency. Competency in something, or competency in doing something, is the ability to do something well [11]. Similarly, the International Board of Standards for Training, Performance, and Instruction defines competency as knowledge, a skill or an attitude that enables one to effectively perform the activities or tasks of a given occupation or function to expected employment standards [12]. In developing an instrument to measure preparedness for e-learning, experts refer to the corresponding competency categories, or constructs, as the dimensions of preparedness. Gulbahar and Kalelioglu [13] defined e-instructors as instructors who might well be very experienced in teaching–learning contexts and even possess a high level of technology literacy. Being an experienced instructor and possessing advanced skills of using technology is necessary but not enough to lead to an instructor becoming an effective e-instructor.

Denis et al. [14] explain that competencies could be categorised as pedagogical, communicational, subject expertise and technological. Klein et al. [15] explain that the categories are professional foundations (communications, professional development, law and ethics, and credibility), planning and preparation, instructional methods and strategies (motivating, presenting, facilitating, questioning, clarifying and correcting, skill retention and transfer), assessment and evaluation, and management (managing the environment and

managing the appropriate use of technology). Lynch and Smith [16] categorise competencies as personal, pedagogical, technical, managerial and institutional.

The dimensions of university lecturers' (faculties') preparedness are linked to how they view their functions. Guasch et al. [17] identify the following specifications for every lecturer' functions/roles: (a) design/planning function, (b) social function to build a positive environment during the teaching/learning process in a virtual environment, (c) instructive function in their roles as facilitators and subject experts, (d) technological domain and (e) management domain. Therefore, lecturers require knowledge and skills to present content and facilitate learning by using technological tools and resources.

Research on how lecturers view e-learning readiness was carried out by Nwagwu [18]. With 240 lecturers from Nigerian universities as respondents, the study concluded that according to the opinion of lecturers, the readiness of society, funding, training, ICT-equipment, and e-learning content development were significant influencers on the readiness of Nigerian universities towards the adoption of e-learning [18]. The readiness of students and human resources were not found to be significant factors towards the adoption of e-learning.

Competent e-instructors are key to successful e-learning implementations and they should have the appropriate skills and experience for the effective implementation of e-learning and blended learning Gulbahar and Kalelioglu [13]. The study concluded that being an experienced instructor and possessing advanced skills of using technology is necessary but not enough to lead to an instructor becoming an effective e-instructor.

Ochogo et al. [19] examined the influence of lecturers' computing competence and preparedness for electronic learning (e-learning). The aim of the study was to investigate the influence of institutional support through providing training programs and funding on lecturers' preparedness to teach in an e-learning environment at the University of Nairobi. The study found no significant relationship between lecturers' preparedness for e-learning and the perceived effectiveness of the existing training program. Lecturers' preparedness was significantly influenced by training in software tools.

The COVID-19 pandemic has caused a rapid transition to online education around the globe. The adoption of e-learning systems during the pandemic is a difficult and challenging process [20], and will continue after the pandemic [21]. This emergency transition to e-learning and faculty development is different from regular transition that requires global collaboration, such as sharing published material [21]. Transition to e-learning is the whole process of change, the actual conversion of each course in an institution, including the training of faculty, and the faculty finalizing their courses and then migrating to the new online environment [22]. The author reported that even in normal condition, transition from face to face to e-learning is considerably time consuming and changes the faculty's role and teaching responsibilities [22].

Alqahtani et al. [20] investigated critical success factors for e-learning during COVID-19. They concluded that technology management, support from management, increased student awareness to use e-learning systems, and demanding a high level of information technology from instructors, students, and universities are the five most influential factors [20]. They highlight that the leading factor for improving the educational process during the pandemic is readiness for e-learning implementation, not how advanced the technology is.

## 3. The Method

### 3.1. Research Approach and Context

This research used a mixed-method approach—it collected and analysed both quantitative and qualitative data from respondents. Considering the scope, this research applied a case study design where lecturers from Indonesia were representative of the population and selected participants with varied backgrounds and experiences. Quantitative data were taken from Likert-based instruments to measure the respondent's perceptions by using deductive, logical thinking, and qualitative data were coded to discover their patterns. They

were then interpreted using deductive, logical thinking. In line with research mapping, as initiated by Saunders et al. [23], this research adopted pragmatism as a paradigm where qualitative and quantitative research should be compared to extract more useful information. It accommodates the more holistic data findings and triangulation since quantitative measurement would be completed alongside in-depth opinion as confirmation. The mixed-method approach has been satisfactorily practiced in education-based research [24–28]. Therefore, the collected and analysed data can be more reliable and qualified.

### 3.2. Participants

This research used voluntary and convenience sampling techniques. These techniques were chosen due to their simplicity, low cost and time investment, and the vast population available. This research focuses on instructors from universities in Indonesia, who hosted the e-learning classes. Relying on social media, this research petitioned participants during April 2020, when the pandemic began in Indonesia, by which time all universities had decided to host online learning. Table 1 summarises their demographic profiles. Jabodetabek is a term for Jakarta (the capital city in Java) and the surrounding cities, Bogor, Depok, Tangerang and Bekasi.

**Table 1.** Respondent Lecturer Demography.

| Attribute | Category | N | Percentage |
|---|---|---|---|
| Location | Java Jabodetabek (the capital and adjacent cities) | 10 | 9 |
| | Java non-Jabodetabek | 55 | 49 |
| | Sumatra | 19 | 17 |
| | Sulawesi | 19 | 17 |
| | Other | 7 | 6 |
| Discipline | Social and Humanities | 20 | 18 |
| | Engineering | 73 | 65 |
| | Education | 10 | 9 |
| | Health | 5 | 4 |
| | Religion | 4 | 4 |
| E-learning Experience | Yes | 71 | 63 |
| | No | 41 | 37 |

This research is confident that the demography is adequate and representative due to the large number of participants and good distribution. Most regions in Indonesia were captured, predominantly Java, which reflects the proportion of universities in Indonesia. Five disciplines are represented, with the majority of participants in engineering. Although 37% of respondents had no prior experience of teaching in an e-learning environment, this research leverages their perceptions to unveil instructors' readiness to teach in an e-learning environment for the first time.

### 3.3. Research Phases

The study was conducted in the following phases: literature search and review, problem formulation, research questions formulation, data collection and analysis, presentation of results and interpretation, and conclusion writing. Literature search and review were carried out using relevant digital libraries (e.g., ACM, Science Direct, and IEEE Xplore) and journals (e.g., Education and Information Technologies) related to online learning and online preparedness. The problem was formulated by reflecting on the researchers' experiences while facilitating online learning, specifically before and during the pandemic, and reviewing the literature. Based on the problem formulation, the research questions were proposed. Data collection was conducted online, and data analyses was conducted for both quantitative and qualitative data.

*3.4. Research Instrument*

This research relied on two basic instruments to measure the readiness of lecturers. It adapted the University of Toledo's [29] instructor readiness questionnaire and made appropriate adjustments to improve quantitative reliability. The study is comprised of four parts (dimensions): Basic Technical Skill; LMS Experience; Course Planning, Time Management, and Communication; Course Design. Each part has four items, except for the last, which has five items.

The instruments were delivered in questionnaires, using five-point Likert scales. The higher the scale, the more strongly the respondents agree with the statement (item). Descriptive statistics were derived from interpreting the collected data by calculating the average and deviation standard. Next, the qualitative data coding process was coded to find themes that reveal the lecturers' perspectives on challenges, motivation, instructional design, collaboration, teaching and learning strategies, available IT infrastructure and other potential themes. The questions are the following:

- Describe what you think, or feel, about your capacity as an educator during this pandemic.
- In responding to the current pandemic, please state three things about, or adjustments to, your teaching strategy.
- Name three online teaching challenges that you experienced.
- As a lecturer, what are your expectations of students while teaching during the pandemic?
- As a lecturer, what are your expectations from the management of the study program, or faculty, while teaching during the pandemic?

## 4. Results and Discussion

The following sections present the research findings to answer two research questions related to lecturers' readiness levels to conduct online classes during a pandemic and lecturers' expectations when participating in online distance education.

*4.1. Instrument Reliability and Validity Tests*

The reliability test checked the consistency of items after repeated trials. In comparison, the validity test was applied to test the validity of a questionnaire in the population. The reliability and validity tests were carried out using SPSS Version 24 software, and the reliability of each part of the questionnaire was tested first, then followed up by confirming the validity of the items in the corresponding part.

Table 2 shows the reliability and validity of the test results. It covers five metrics: Cronbach's Alpha (CA), Cronbach's Alpha Based on Standardized Items (CASI), the smallest Corrected Item-Total Correlation (CITC), the smallest Cronbach's Alpha if Item Deleted (CAID) and R-Table. The CA value ranges from 0 to 1, with higher values indicating greater internal consistency. A CA of 0.7 or higher is considered reliable. If the CITC and CAID values are higher than the R-Table value, each item in the questionnaire is reliable and valid.

**Table 2.** Reliability and Validity Tests of the Lecturer Readiness Questionnaire.

| Part | CA [1] | CASI [2] | Lowest CITC [3] | Lowest CAID [4] | R-Table | Conclusion |
|---|---|---|---|---|---|---|
| Part A: Basic Technical Skills | 0.888 | 0.897 | 0.705 | 0.717 | 0.1857 | Reliable and valid |
| Part B: LMS Experience | 0.844 | 0.847 | 0.619 | 0.766 | 0.1857 | Reliable and valid |
| Parts C and D: Course Planning, Time Management and Communication | 0.722 | 0.732 | 0.339 | 0.650 | 0.1857 | Reliable and valid |
| Part E: Course Design | 0.794 | 0.797 | 0.501 | 0.734 | 0.1857 | Reliable and valid |
| Part A: Basic Technical Skills | 0.888 | 0.897 | 0.705 | 0.717 | 0.1857 | Reliable and valid |

[1]—CA: Cronbach's Alpha; [2]—CASI: Cronbach's Alpha Based on Standardized Items; [3]—CITC: Corrected Item-Total Correlation; [4]—CAID: Cronbach's Alpha if Item Deleted

This research adopted five parts/dimensions of the instructor readiness assessment. The lecturer readiness questionnaire's reliability and validity test showed that parts A, B, and E are valid and reliable. However, parts C and D are not reliable because the CA values for parts C and D are 0.568 and 0.542, respectively, less than the accepted lower limit of 0.7. However, parts C and D are valid because for each item the value of CITC and CAID is greater than the R-Table value. This research found that PTC-06 (I feel more comfortable communicating through speech than through writing) is invalid and should be deleted. After combining parts C and D, and deleting the PTC-06 item, they become reliable and valid, and the analysis is based on this revised version. Table 2 shows a summary of the reliability and validity tests.

### 4.2. Quantitative Interpretation of Lecturers' Perspectives

This section elaborates on and discusses the findings related to the lecturers' perspectives on their readiness to teach online.

#### 4.2.1. Correlation Analysis

By using Chi-square correlation analysis, the lecturers' readiness to teach online is closely related to the lecturers' level of convenience using the LMS to design online classes, facilitate students in the learning process and the lecturers' ability to communicate well through writing (LMS-01, LMS-02, PTC-02 and PTC-05).

#### 4.2.2. Correspondence Analysis

Correspondence plots are carried out for each aspect of the question to obtain more accurate results. Below is a correspondence plot. To simplify the plot, item names are shortened to "A" for "BAS", "B" for "LMS", "C" for "PTC" and "D" for "DSG".

Based on the correspondence plots in Figures 1–4, it is concluded that lecturers who have taught online are well prepared to use the e-learning system, which includes time management, class planning and online class design. However, lecturers who previously only taught face-to-face (do not have online teaching experience) are less prepared for class planning, time management and online communication. Clustering (k-means) was applied to see the characteristics of lecturers in teaching online and Table 3 summarises the results.

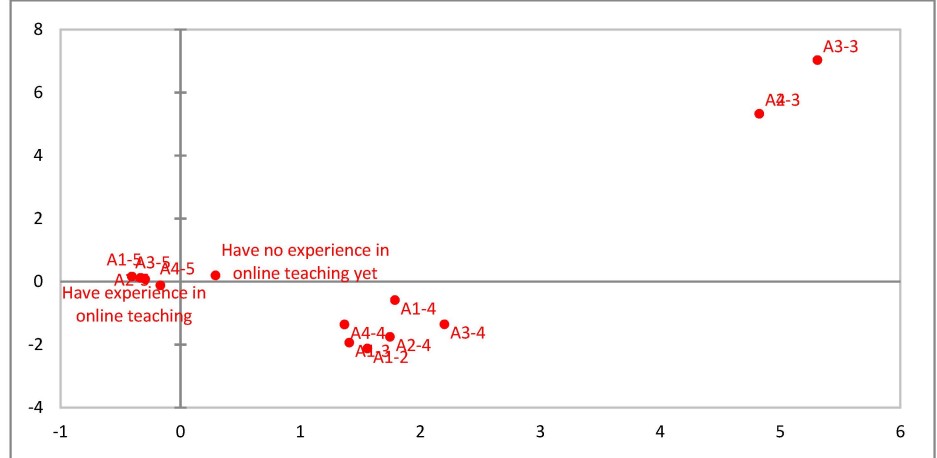

**Figure 1.** A correspondence plot of Part A (Basic Technical Skill).

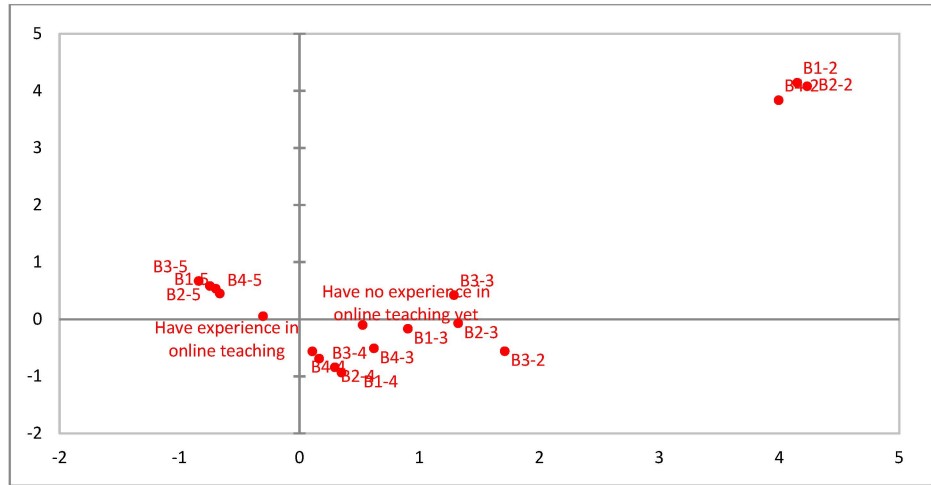

**Figure 2.** A correspondence plot of Part B (LMS Experience).

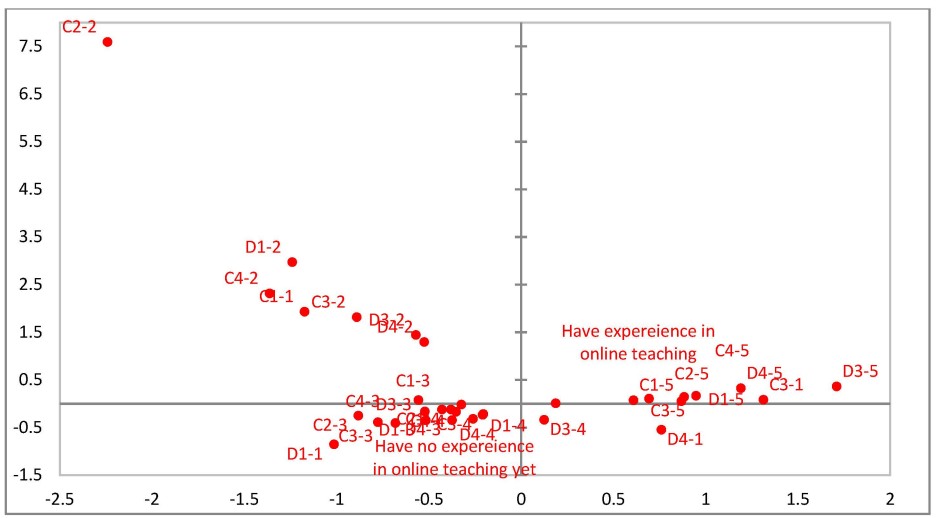

**Figure 3.** A correspondence plot of Parts C and D (Course Planning, Time Management and Communication).

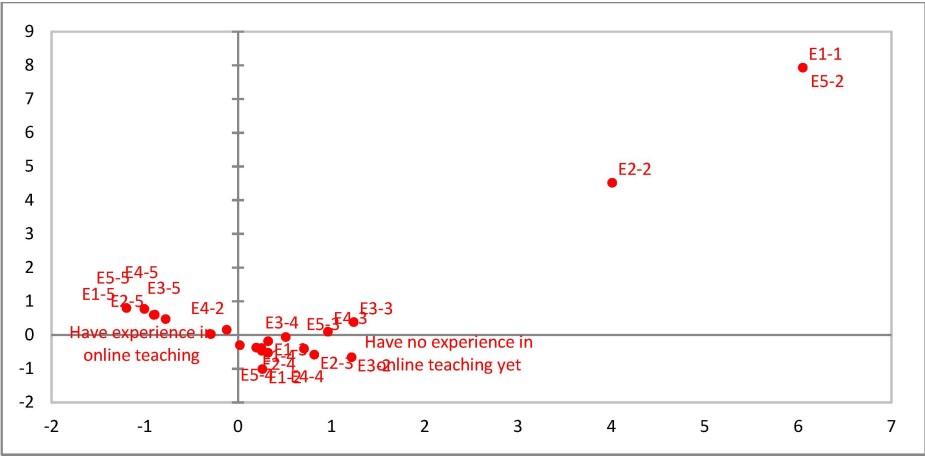

**Figure 4.** A correspondence plot of Part E (Course Design).

**Table 3.** Comparison between Lecturer's Clusters.

| Issue | Lecturers Who Are Prepared for Online Teaching | Lecturers Who Are Less Prepared for Online Teaching |
| --- | --- | --- |
| Have previous online teaching experience | Yes | No |
| Basic skills in operating electronic devices and LMS | Have good basic skills to operate electronic devices and LMS | Have basic skills to operate electronic devices (such as managing files and using browsers) |
| LMS usage | Already feel comfortable | Not ready |
| Class management and design | Have good skills to design classes and time management of the student learning process | Less prepared to design online classes and manage time. |
| Communication style | Capable of communicating online, both verbally and in writing (such as conveying feelings/affections) | Less ability to communicate via text or audio/video devices |

Aspects that need to be considered so that lecturers are better prepared for online teaching are the ability to use the LMS correctly and express feelings/affections through writing, audio or video, which are weak points due to a lack of previous online teaching experience. Table 4 shows the complete scoring for each instrument.

**Table 4.** Faculty Online Teaching Readiness Survey.

| Code | Indicator | SD | Mean | Cluster 1 | Cluster 2 |
| --- | --- | --- | --- | --- | --- |
| Do you have pre-pandemic online teaching experience? | | | | Yes | No |
| Basic Technical Skills (average 4.835) | | | | | |
| **BAS-01** | I can use office applications, such as Open Office, Microsoft Word and Microsoft PowerPoint. | 0.492 | 4.786 | 5 | 5 |
| **BAS-02** | I can perform file management on my computers, such as copying, moving, renaming and deleting files or folders. | 0.407 | 4.848 | 5 | 5 |
| **BAS-03** | I can send and receive emails and open and send email attachments. | 0.349 | 4.884 | 5 | 5 |
| **BAS-04** | I can use an Internet browser, such as Google Chrome, Firefox or Safari to search the Web and upload/download files and programmes. | 0.429 | 4.821 | 5 | 5 |
| LMS Experience (average 4.230) | | | | | |
| **LMS-01** | I feel comfortable using an LMS (such as Moodle and Google Classroom) to build an online course. | 0.824 | 4.295 | 5 | 4 |
| **LMS-02** | I feel comfortable using features in the LMS to facilitate student learning. | 0.748 | 4.313 | 5 | 4 |
| **LMS-03** | I feel comfortable using LMS assessment tools to evaluate student performance. | 0.738 | 4.179 | 5 | 4 |
| **LMS-04** | I feel comfortable using the LMS to record student grades. | 0.833 | 4.134 | 5 | 4 |
| Course Planning, Time Management and Communication (average 3.980) | | | | | |
| **PTC-01** | I am detail oriented. | 0.725 | 4.277 | 5 | 4 |
| **PTC-02** | I am good at organising teaching materials. | 0.691 | 4.250 | 5 | 4 |
| **PTC-03** | I expect online teaching to take more time than face-to-face instruction, and I am prepared for it. | 1.009 | 3.991 | 4 | 4 |
| **PTC-04** | I am willing to provide timely and constructive feedback on student performance. | 0.774 | 4.152 | 4 | 4 |
| **PTC-05** | I feel comfortable communicating through writing and can do it easily. | 0.902 | 3.920 | 4 | 4 |
| **PTC-06** | I feel more comfortable communicating through speech than through writing. | 0.843 | 4.027 | - | - |
| **PTC-07** | I feel comfortable conveying my personality and/or emotions through writing. | 0.849 | 3.518 | 4 | 3 |
| **PTC-08** | I feel comfortable conveying my personality and/or emotions through speaking (audio/video). | 0.965 | 3.705 | 4 | 3 |
| Course Design (average 4.082) | | | | | |
| **DSG-01** | I feel comfortable writing measurable learning objectives based on Bloom's taxonomy. | 0.781 | 3.857 | 4 | 4 |
| **DSG-02** | I feel comfortable designing active learning activities that allow students to interact with their peers, instructors and course content. | 0.646 | 4.205 | 4 | 4 |
| **DSG-03** | I understand copyright law and fair use guidelines when using copyrighted materials. | 0.741 | 4.250 | 4 | 4 |
| **DSG-04** | I understand accessibility policies on student needs. | 0.737 | 4.125 | 4 | 4 |
| **DSG-05** | I know how to accommodate student needs. | 0.716 | 3.973 | 4 | 4 |

### 4.2.3. Part A: Basic Technical Skill

Generally, this dimension was the best from the lecturer's perspectives. It indicates that lecturers have strong basic technical skills as a baseline to use e-learning platforms for online courses. Their strengths are shown by the five items as the most elected option, while the standard deviation was relatively low. Achievement in this dimension also became a meaningful foundation to encourage lecturers' digital literacy. This research did not find critically bad areas of concern since almost all lecturers have adequate basic technical skills, and most Indonesian people use mobile phones.

### 4.2.4. Part B: LMS Experience

All indicators have a score of 4.00 or more. This shows that lecturers are able to use LMS, with fast adaptation. Interestingly, some respondents claimed that they had no experience of online classes before the pandemic. This research argues that LMS platforms have good usability so that lecturers feel easy, comfortable and satisfied when using them. Moreover, most lecturers state their conformity when leveraging LMS as an assessment medium, not only for teaching agendas. In an open-question answer, out of 112 respondents, 12 and seven people out of 112 respondents stated that they used Zoom and Google Classroom as the LMS platforms, respectively.

### 4.2.5. Parts C and D: Course Planning, Time Management and Communication

Generally, this research captured a balance score distribution between 4 and 5. This research emphasis became a crucial issue since communication determines whether, and how, lecturers transfer knowledge, skills and inspire their students. This research argues that the pandemic occurred mid-semester, while the planning was done before the semester. Therefore, most lecturers can focus on migrating their course agenda from a physical class to an online class, but this research also highlights that the pandemic was unexpected, so most lecturers had a tactical solution by hosting online classes with low feasibility, and they did not have a strategic solution.

Statistically, item PTC-07 was the most significant challenge and is the lowest indicator (3.52). It shows that many lecturers stated their inability to express themselves through written media. This feeling became a significant challenge due to the large differentiation between communication styles in a physical class and an online class. In the physical classroom, the interaction between lecturer and student occurs by combining audio, visual and kinaesthetic methods, but online classes rely on written communication to minimise data transmissions since video generates much more data. Unfortunately, written messages can obscure real emotional feelings, such as when the lecturer expresses appreciation or disagreement.

Interestingly, an item on the readiness to spare the time to teach online had a score of 3.99. This indicates critical issues concerning the online class paradigm. This research identified that most lecturers thought that online classes required more time to compose a video storyboard, upload course material, decompose the online course scenario, monitor student activity and review the assignment. As stated by RL-29, a lecturer needed more time to determine the best method to assess student achievement. A similar complaint was expressed by RL-106 who felt confused about how to objectively measure student achievement. Several respondents revealed they had decided to revise course planning by switching from lectures to assignments during the pandemic.

Item PTC-08 (I feel comfortable conveying my personality and/or emotions through speaking (audio/video)) confirms the above-mentioned issues—it scored a relatively low 3.705 when compared with the other items. In contrast, respondents claimed that communication through written media was relatively easy, as captured in PTC-05 (I feel comfortable communicating through writing and can do it easily) with a score of 3.920. This research concludes that lecturers experienced difficulties in carrying out easy tasks. This could be due to the rapid transition to online as a result of the unexpected pandemic. Internet access is another challenge since some cities have poor internet connectivity, according to RL-81

and RL-86. This pushed the lecturers to choose synchronous online classes, while online classes relied on asynchronous internet access due to its instability.

### 4.2.6. Part E: Course Design

This dimension had an average score of 4.08, which reflects several challenging issues. First, this research found that lecturers' conformity to teach online was not optimised since they did not fully believe the learning goals could be achieved, which relates strongly to the third dimension. At the beginning of the semester, lecturers had set learning outcomes as a standardised goal that should be cascaded into the syllabus. Unfortunately, their designed syllabus had to be revised, especially the content, delivery and other course design attributes. This can lower lecturers' confidence about the previous syllabus and the achievement of its learning goals as well.

Fortunately, many elective LMS platforms can be leveraged to host online classes. They provide many useful features that accommodate lecturers' course design requirements. For example, some LMS platforms offer a submission menu that simplifies student assignment uploads and downloads by lecturers. As mentioned by RL-32, online classes should provide impetus to explore more advanced features, such as creating online attendance registers, online examinations or file sharing.

### 4.3. Qualitative Interpretation of Lecturers' Perspectives

Using a qualitative approach, this research captured lecturers' perceptions through open-ended questions embedded in the survey, after the quantitative instruments. The results are presented in the following table.

This research captured meaningful statements by lecturers by using codification. Tagging was used to count, and cluster responses based on their similarity. Based on the results of the thematic coding, six themes are the most dominant. If a theme had a greater frequency, more lecturers had similar perceptions, making it a more essential issue. Table 5 shows the codification summary.

**Table 5.** The Most Dominant Challenges Faced by Lecturers.

| Theme of Challenges | Frequency | Example of Responses |
|---|---|---|
| ***Internet connection and internet fee (quota)*** The biggest challenge was the unstable internet connection. More than 55% of lecturers mentioned unstable internet access, especially experienced by students who live in remote areas. Internet connection problems interfere with the teaching and learning process. Lack of equipment support was also an obstacle for some students. More than 23% thought that online learning disadvantages less fortunate students due to internet quotas. | 55.36% | "The internet connection was poor, students have network access constraints so they cannot attend lectures, and the quota was limited." "Lack of equipment for underprivileged students." "The internet quota for students is limited, especially those who live in rural areas where the network/signal is sometimes slow, thus limiting video conferencing." "Additional fees for internet quota." |
| ***Course delivery and teaching strategies*** More than 23% of respondents acknowledged the challenges of delivering effective, creative, and relevant material and matching subject characteristics so that they were easy to understand. Lecturers recognised that online learning requires different teaching skills. | 32.2% | "Must carefully explain so that it is more effective and easier to digest by students." "Creativity in delivering relevant material." "Teaching online is different from face to face, more difficult and requires high commitment." "Less optimal for lesson that require practice in the laboratory." "To create and describe the formula formulas and their applications are rather complicated." |

**Table 5.** *Cont.*

| Theme of Challenges | Frequency | Example of Responses |
|---|---|---|
| *Evaluation*<br>Some lecturers experienced serious challenges in evaluating learning outcomes and processes in the four most dominant aspects: an exam model that measures understanding well; administering and monitoring learning progress; encouraging students to maintain integrity and honesty; and monitoring the assessment process to avoid cheating. | 16.96% | "Still looking for an evaluation method that truly describes the abilities of students."<br>"More difficult to check and provide feedback on student work."<br>"Proper administration of exams, exam models."<br>"Difficult to control the student working process, whether doing it themselves or cheating." |
| *Time constraints*<br>Compared with the setting before the Covid-19 pandemic, lecturers felt that it took longer to prepare lecture materials. They admitted that they were constrained by having to manage their time to adapt to the new teaching modes. | 11.6% | "I need more time to prepare lecture materials so that the objectives and learning goals are conveyed by students even though the limitation of non-verbal communication."<br>"It is difficult to manage time, during WFH . . . need time to adapt." |
| *Monitoring*<br>Lecturers found it challenging to ascertain whether the learning process occurs, monitor understanding and control whether tasks completed by the student or by someone else. | 9.8% | "Difficult to control whether students do their work or copy someone else's work."<br>"Cannot be monitored whether students are involved in the learning process or not."<br>"Still difficult to assess the level of understanding in discussion forums." |
| *Motivating students*<br>Lecturers admitted they were challenged in helping to improve students' readiness to undergo online learning. Lecturers were challenged in motivating students to focus on, and being, actively involved in the learning process. Lecturers saw the gap in student readiness. | 9.8% | "Provide support and enthusiasm to learn online, overcome boredom, maintain student focus."<br>"Difficult to make students learn actively, through discussion."<br>"Not all students are ready for online lectures." |

Lecturers' challenges can be divided into two dimensions: (1) the unstable internet connection and additional expenses for internet usage that burden students (55.36%) and (2) lecturers are more challenged in carrying out their roles (71.16%). Lecturers are challenged in preparing teaching materials, delivering the courses, monitoring student progress and engagement, evaluating learning and helping students maintain motivation and engagement, which forces them to invest more time and effort.

## 5. Implications

### 5.1. Theoretical Implications

This research has revealed empirical results on lecturers' readiness for teaching online courses. Interesting facts were found that have theoretical implications. Lecturers' readiness was relatively high (4 to 5 on the scale) although they had a medium level of experience in e-learning (63%). This implies that experience does not automatically make someone ready to conduct online courses. This research found that most of the lecturers received training to host online classes. This opportunity accelerated their knowledge and skills to allow them to be better prepared when running online classes during the pandemic. This situation reflects a study conducted by Reyes-Chua et al. [6] who said that a lack of faculty member training to use e-learning classrooms is an essential problem in delivering online courses during the pandemic. E-learning will become a necessity in education. Instructors must increase their capabilities to run e-learning well. Besides being ready to run e-learning, lecturers must also help students to be ready to study in an e-learning environment.

This research also addresses two frequent issues that lecturers face: unstable internet access and self-management. These issues are coherent with students' critical problems, as

mentioned by Ebner et al. [5]. These authors noticed that students could suffer depression and anxiety due to unfavourable study environments at home, which lead to a lack of self-management. Similar to Ebner et al. [5], this research classifies both issues as barriers that should be tracked when assessing students' readiness. It suggests that both these frequent issues become barriers for lecturers and students. Online collaborative learning, for example, using a discussion board, can bring students and lecturers closer, and thus reduce anxiety [5]. Lecturers need training to improve their preparedness to conduct e-learning and to help students become ready to learn in an e-learning environment. Caliskan et al. [30] suggested that universities should have distance education centre to help lecturers tackle technical problems. Internet access has been the most frequent issue raised by lecturers. It indicates the lack of the readiness of ICT infrastructure. This is consistent with the study conducted by Nwagwu (2020), that found ICT-equipment readiness to be one of the most significant factors influencing lecturers' opinions about the readiness of universities to adopt e-learning.

This research actualized a mixed-method approach to enable more holistic findings. It also accommodated the comparison to ensure data reliability and validity, especially in data interpretation. Therefore, this research has proved the mixed-method approach's strength as claimed by [24–26,28].

### 5.2. Practical Implications

This subsection focuses on emphasising the necessary LMS features that should be developed to encourage lecturers' readiness. Lecturers' readiness and confidence to use LMS for online courses will increase by implementing appropriate features. First, lecturers require time-management features. They are physically separated from each other, so they may forget many tasks. When lecturers start their daily activities, they should find their tasks and complete them with less interaction with others. Therefore, their skill to maintain schedules should be improved. Second, this research highlights the importance of notification features that alert lecturers of any updated information in an online course, such as edited assignments, new comments in a forum or submission reminders. This should reduce miscommunication because the lack of information is due to poor LMS design. Third, statistics tracking should be developed to measure lecturers' LMS adaptation rate. This will enable the university to adjust the LMS structure and content to enhance readiness. Fourth, all LMS business processes should be measured, including the amount of data transmitted. People usually buy prepaid internet packages, such as 1 GB/month and 50 GB/month in Indonesia. It implies their internet access is limited, so that data transmission during online classes should be minimised to ensure their continuity during online learning. Furthermore, findings in this research should become an essential consideration for university management and government to encourage the quality and effectiveness of online learning during the COVID-19 pandemic. For example, university management can formulate appropriate standards for online learning by accommodating lecturers' readiness and teaching methods.

### 6. Conclusions

This research provides empirical findings on lecturers' readiness to conduct online courses. Most lecturers had adequate readiness to host online classes during the COVID-19 pandemic. The study combined quantitative and qualitative data gathered from university lecturers in Indonesia. For quantitative measurements, this research adopted the instructor readiness questionnaire of the University of Toledo [29] and made appropriate adjustments to improve reliability. It comprised four parts (dimensions): (1) Basic Technical Skill; (2) LMS Experience; (3) Course Planning, Time Management and Communication; (4) Course Design. Using descriptive statistics, their scores (out of 5.00) were 4.834, 4.835, 4.230, 3.980 and 4.082, respectively. Lecturers' readiness was relatively high (4 to 5 on the scale) although they had a medium level of experience in e-learning (63%) before the

pandemic that accelerated their knowledge and skill, allowing them to be better prepared when running online classes during the pandemic.

This research also highlights two frequent issues that lecturers face: unstable internet access and self-management. It suggests that these two issues become barriers for lecturers and students. This research provides several solutions to overcome these issues by proposing features in LMS. With appropriate features, lecturers will be better prepared and more confident when using LMS for online courses.

## 7. Outlook for Future Research

After performing a quantitative and qualitative assessment on lecturers' readiness, this research makes several recommendations for future research. First, this research suggests using a broader and more diverse sample to provide a more holistic view of lecturers in Indonesia, especially lecturers with new experiences. To obtain a broader and more diverse sample, this research proposes snowball and purposive sampling techniques. These techniques can be used by associations of lecturers or social networking to reach greater potential populations.

Second, this research captures lecturers' views about unstable internet access. Interestingly, internet access was not an instrument in lecturers' readiness since the instruments were created in European countries with stable internet access. Therefore, this research suggests that future research should adjust the readiness model and instrument in line with Information Technology (IT) infrastructure, such as the university's IT service and internet access. Moreover, lecturers and students are in separate locations, so their interaction was influenced by IT infrastructure.

**Author Contributions:** Conceptualization, K.J. and H.B.S.; methodology, H.B.S.; software, A.G.; validation, T.S., P.O.H.P.; formal analysis, H.B.S.; investigation, T.S.; resources, A.G., data curation, T.S.; writing—original draft preparation, K.J.; writing—review and editing, H.B.S.; visualization, P.O.H.P.; supervision, K.J.; project administration, A.G.; funding acquisition, K.J. All authors have read and agreed to the published version of the manuscript.

**Funding:** This research was funded by The Research Grant: Publikasi Terindeks Internasional (PUTI) Q2 2020, Number: NKB-1480/UN2.RST/HKP.05.00/2020 funded by DRPM Universitas Indonesia.

**Institutional Review Board Statement:** Not applicable.

**Informed Consent Statement:** Informed consent was obtained from all subjects involved in the study.

**Data Availability Statement:** The data presented in this study are available on request from the corresponding author. The data are not publicly available due to the confidential information.

**Conflicts of Interest:** The authors declare no conflict of interest.

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
