# Peer review of "Lecturer Readiness for Online Classes during the Pandemic: A Survey Research"

_education, doi:10.3390/educsci11030139_

Round 1
Reviewer 1 Report
First of all, many thanks to the authors for the opportunity to read this work.
It is a work with an interesting and current theme.
The review of the previous literature is scarce. It only focuses on defining the terms preparation and competence. It would be convenient to expand and focus this section to other aspects related to the research problem, the research objectives or questions.
In the section on study context and research approach, it would be convenient to justify the mixed method used with some bibliographic references that endorse or justify the suitability of this method for the research objectives set out in your work.
In section 4, called "Results & Discussion", the adequate presentation of results is developed. However, it does not establish any discussion about them. Therefore, I ask that the discussion of the results with previous research on the subject under investigation be developed in an adequate way.
Review the structure of section "5. Implications", the titles of subsections 5.1 and 5.2 as they have the same title "Theoretical implications" but their content is confusing.
It would be convenient that in section 5 of implications of the study the novelties and contributions of the same are clear both to the field of research and to the practical application and usefulness of the same in their transfer.
Updating and deepening of bibliographic references is recommended as they are insufficient.
In short, the work needs to reinforce its results with greater academic solidity and support in its arguments.
Author Response
Regarding the reviewer’s comments to our paper entitled “Lecturer Readiness for Online Classes During the Pandemic: A Survey Research” we have addressed the comments as mentioned below.
Reviewer #1
|
|
Comment |
Respond |
Location |
|
1. |
The review of the previous literature is scarce. It only focusses on preparation and competence. It would be convenient to expand and focus to other aspects related to the research problems, questions and objectives. |
Add more literature review
|
Section 1 (Introduction) and Section 2 (Literature Review) |
|
2. |
In the section of the context and study approach, it would be convenient to justify the mixed method used with some bibliographic references that endorse the suitability of this method for research objectives set in your work. |
Add explanation on method and references |
Sub-section 3.1 (Research Approach and Context) |
|
3. |
In Section 4, called “Results & Discussion” the adequate result is developed. However, it does not establish any discussion about it. Therefore, I ask that the discussion of the result |
Add opening paragraph and minor revision |
Section 4 (Results and Discussion) |
|
4. |
Review the structure of Section 5 Implications. The Title of subsections 5.1 and 5.2 are the same but their content is confusing. |
Revise the title |
Sub-section 5.2 (Practical Implications) |
|
5. |
It would be convenient that in Section 5 of Implications of the study the novelties and contributions of the same are clear both to the field of research and to the practical applications and usefulness of the same transfer. |
Add explanation in section 5 (end of section 5.1 and 5.2) |
Sub-section 5.1 (Theoretical Implications) and 5.2 (Practical Implications) |
|
6. |
Updating and deepening the bibliographic references is recommended as they are insufficient. |
Add more literature review - |
Section 2 (Literature Review) and Sub-section 3.1 (Research Approach and Context) |
|
7. |
In short, the work needs to reinforce its results with grater academic solidity and support in these arguments. |
Expand Lit review, mixed methods, results conclusion |
Section 2 (Literature Review), Sub-section 3.1 (Research Approach and Context), Sub-section 5.1 (Theoretical Implications), and 5.2 (Practical Implications) |
Reviewer 2 Report
Add citations to the introduction to support statements and help the reader see connections to the theoretical background and empirical research. For example, statements such as, “many universities use an LMS only as a course-material repository and course reporting system because there is less communication and interaction amongst participants than in a physical class” need citations to support the argument.
Clarify the argument for conducting the study and assessing instructor readings. For example, the following statement is unclear “unfortunately, this policy was implemented without assessing lecturers’ readiness and the government should have been more agile.” It is unclear how the government could have assessed readiness prior to shifting to online due to the pandemic. This happened suddenly, so please explain how the government should have been more agile. Also, provide supporting literature for the argument that readiness is connected to successful implementation.
The definition of the terms in the literature section is helpful. Add literature related to how readiness is connected to successful implementation.
Clarify the purpose of the study.
“This research considers whether lecturers have equal and appropriate readiness.” It is unclear what is meant by equal readiness. Equal to whom? Is the study trying to see if instructors all have the same level of readiness?
“As a practical benefit, this research contributes stakeholders’ input to implement tactical policies that improve teaching staff readiness for distance teaching.” Clarify how the study contributes stakeholders’ input.
Add more detail about the methods of analysis. Also, note section 3.1 and section 3.2 are the same.
Add more detail to clarify findings in section 4.2.3. “This research did not find critically bad areas of concern since almost all lecturers have adequate basic technical skills, and most Indonesian people use mobile phones.” Explain what is meant by “critically bad areas of concern.” I did not see a question in the survey that asked about mobile phone use.
In section 4.2.3 “This dimensions findings are positive because all indicators have a score of 4.00 or more. It shows the lecturers are able to use LMS, with fast adaptation.” It is unclear what is meant by fast adaptation.
“In an open-ended question answer, several respondents mention Zoom and Google Classroom as LMS platforms used by 12 and 7 respondents, respectively.” Clarify how many respondents answered the open-ended questions in total.
Section 4.2.5 needs more detail and clarity. It is unclear how the data supports the following statements: “This research argues that the pandemic occurred mid-semester, while the planning was done before the semester. Therefore, most lecturers can focus on migrating their course agenda from a physical class to an online class, but this research also highlights that the pandemic was unexpected, so most lecturers had a tactical solution by hosting online classes with low feasibility, and they did not have a strategic solution.”
Section 4.3 – it seems the information provided in this section (questions asked) could be part of the methods section instead of the findings.
Table 5, challenge: Course delivery and teaching strategies: “Lecturers recognised that online learning requires different teaching skills” – Are there quotes that could be selected to help illustrate this point?
Table 5, Evaluation: “Some lecturers experienced significant challenges…” What do you mean by significant? Does this refer to the frequency or something different. Clarify what is significant.
Table 5, Time Constraints: “Compared with the setting before the COVID-19 pandemic, lecturers felt that it took longer to prepare lecture materials.” Explain how the study compare to the pre-COVID setting?
5.1 Theoretical Implications: “This research found that most of the lecturers received training to hos online classes.” Provide more detail about the training provided. Was the training pre-covid or during the transition from face-to-face to online teaching? It is unclear how this was found in the research.
Clarify what part of the data supported the following claim: “Online learning requires independent learning and self-motivation, and students and lecturers might feel isolated. Students might also feel constrained to seek help when they need to.”
Add more detail and connection to the data:
- “Online collaborative learning, for example, using a discussion board, can bring students and lecturers closer, and thus reduce anxiety.”
- “Although internet access has been the most frequent issue, quantitative instruments from the University of Toledo do not cover it.”
- “Therefore, the instruments should be enriched by assessing digital infrastructure, such as internet access, storage size and power source stability.”
- “Fourth, all LMS business processes should be measured, especially the spent data package to accommodate lecturers with limited internet access.” Clarify “spent data package.”
Author Response
Regarding the reviewer’s comments to our paper entitled “Lecturer Readiness for Online Classes During the Pandemic: A Survey Research” we have addressed the comments as mentioned below.
Reviewer #2
|
No |
Comment |
Respond |
Location |
|
1.
|
Add citations to the introduction to support statements and help the reader see the connection the theoretical background and empirical research. For example, statements such as “many universities use an LMS only as a course-material repository and course reporting system because there is less communication and interaction amongst participants than in a physical class” need citation to support the argument. |
Add more citations. |
Section 1 (Introduction), Section 2 (Literature Review), and Sub-section 3.1 (Research Approach and Context) |
|
2. |
Clarify the argument for conducting the study and assessing instructors’ readings. For example, the following statemen is unclear “unfortunately, this policy was implemented without assessing lecturers’ readiness and the government should be more agile.” It is unclear how the government could have assessed readiness prior to shifting to online due to the pandemic. This happened suddenly, so please explain how the government should be more agile. Also, provide supporting literature for the government readiness is connected to successful implementation. |
Change the phrase.
|
Section 1 (Introduction) |
|
3. |
The definition of the terms in the literature section is helpful. Add literature related to how readiness is connected to successful implementation. |
See response to Rev#1 question 2 |
Sub-section 3.1 (Research Approach and Context) |
|
4. |
Clarify the purpose of the study. |
Add the purpose of the study in the abstract. |
Abstract and Section 1 (Introduction) |
|
5. |
“This research considers whether lecturers have equal and appropriate readiness.” It is unclear what is meant by equal readiness. Equal to whom? Is the study trying to see if the instructors all have the same level of readiness? |
Change the phrase “all lecturers have appropriate readiness” |
Section 1 (Introduction) |
|
6. |
“As a practical benefit, this research contributes stakeholders’ input to implement tactical policies that improve teaching staff readiness for distance learning.” Clarify how the study contributes stakeholders’ input |
Change the phrase to the more clear and appropriate meaning. Here, ‘input’ mean information to be considered. |
Section 1 (introduction) |
|
7. |
Add more detail about the methods of analysis. Also, note section 3.1 and section 3.2 are the same. |
Add more explanation on the method. |
Sub-section 3.1 (Research Approach and Context) |
|
8. |
Add more detail to clarify findings in section 4.2.3 “these dimensions findings are positive because all indictors have a score of 4.00 or more. It shows that the lecturers are able to use LMS, with fast adaptation” |
Delete “these dimensions findings are positive because” without changing the thesis sentence. |
Sub-section 4.2.3 (Part A: Basic Technical Skill) |
|
9. |
“In open-ended question answer, several respondents mention Zoom and Google Classroom as LMS used by 12 and 7 respondents respectively.” Clarify how many respondents answered the open-ended questions in total. |
Add “out of 112 respondents, 12 and seven people out of 112 respondents stated that they used Zoom and Google Classroom as the LMS platforms “ |
Sub-section (4.2.4. Part B: LMS Experience) |
|
10. |
Section 4.2.5 needs more detail and clarity. It is unclear how the data supports the following statements: “This research argues that the pandemic occurred mid-semester, while the planning was done before the semester. Therefore, most lecturers can focus on migrating their course agenda from physical class to an online class, but this research also highlights that the pandemic was unexpected, so most lecturers have tactical solution by hosting online classes with low feasibility, and they did not have a strategic solution.” |
We have updated: “This research argues that the pandemic occurred mid-semester, while the planning was done before the semester. Therefore, most lecturers can focus on migrating their course agenda from physical class to an online class, but this research also highlights that the pandemic was unexpected, so most lecturers have tactical solution by hosting online classes based on their readiness level and their past experiences.”
|
Sub-section 4.2.5 (Parts C and D: Course Planning, Time Management and Communication) |
|
11. |
Section 4.3 – it seems the information provided in this section (question asked) could be part of the methods instead of the findings. |
Move to ‘Methods’ section |
Section 3 (The Methods) |
|
12. |
Table 5, challenge: “Some lecturers experienced significant challenges ..” what dou you mean by significant? Does it refer to frequency or something different. Clarify what is significant. |
Change the word “significant” à “serious” |
Table 5 |
|
13. |
Table 5, Time Constraint: “Compare with the setting before the COVID-19 pandemic, lecturers felt that it took longer to prepare lecture material”. Explain how the study compare to the previous COVID setting? |
This is based on lecturers’ response (see the next column). The study does not compare the two settings. |
Table 5 |
|
14. |
5.1 Theoretical Implications “This research found that most of the lecturers received training to host online classes. “Provide more detail about the training provided. Was the training pre-covid or during the transition from face-to-face to online teaching? Is is unclear how this was found in the research?
|
We have updated:
"This research found that most of the lecturers received training to host online classes. Some large public and private universities have teaching and learning development units that host the lecturers' capacity building events to enhance lecturers' skills in teaching online classes. While some of the events are conducted internally for their staffs, some others are opened for public." |
Sub-section 5.1 (Theoretical Implications) |
|
15. |
Clarify what part of the data supporting the following claim: ”Online learning requires independent learning and self-motivation, and students and lecturers might feel isolated. Students might also constrained to seek help when they need to. |
Delete: authors’ opinion based on experiences |
Sub-section 4.3. (Qualitative interpretation of lecturers’ perspectives) |
|
16. |
Add more detail and connection to the data: “online collaborative learning, for example, using a discussion board, can bring students and lecturers closer, and thus reduce anxiety.” |
Add reference “[5] “ |
Sub-section 5.1 (Theoretical Implications) |
|
17. |
“Although internet access has been the most frequent issue, quantitative instruments from the University of Toledo do not cover it.”
Therefore, the instrument should be enriched by assessing digital infrastructure, such as internet access, storage size and power source stability.” |
Revise ICT-infrastructure readiness (not lecturers’ readiness) Internet access has been the most frequent issue raised by lecturers. It indicates the lack of the readiness of ICT infrastructure. This consistent with the study conducted by Nwagwu (2020), one of significant factors influencing lecturers’ opinions about readiness universities to adopt e-learning ICT-equipment readiness. |
Sub-section 5.1 (Theoretical Implications) |
|
18. |
Fourth, all LMS business process should be measured, especially the spent data package to accommodate lecturers with limited internet access. “Clarify “spent data package” |
Add explanation about ‘data package’ tant applied in Indonesian context |
Sub-section 5.1 (Theoretical Implications) and 5.2 (Practical Implications) |
Round 2
Reviewer 1 Report
Tras la inclusión de las modificaciones propuestas, el artículo se ha mejorado considerablemente y, por tanto, está listo para su publicación.
Felicitaciones a los autores por su trabajo.
Reviewer 2 Report
I reviewed the revisions and believe the manuscript has been sufficiently updated to now warrant publication. Some minor copyediting could help refine the article prior to publication.